# Non-Thermal Plasma Application in Medicine—Focus on Reactive Species Involvement

**DOI:** 10.3390/ijms241612667

**Published:** 2023-08-11

**Authors:** Julia Moszczyńska, Katarzyna Roszek, Marek Wiśniewski

**Affiliations:** 1Department of Materials Chemistry, Adsorption and Catalysis, Faculty of Chemistry, Nicolaus Copernicus University in Toruń, Gagarina 7, 87-100 Toruń, Poland; jmoszczynska01@gmail.com; 2Department of Biochemistry, Faculty of Biological and Veterinary Sciences, Nicolaus Copernicus University in Toruń, Lwowska 1, 87-100 Toruń, Poland; kroszek@umk.pl

**Keywords:** non-thermal plasma, medical applications, wound healing, cancer treatment, plasma-based therapies, sterilization

## Abstract

Non-thermal plasma (NTP) application in medicine is a dynamically developing interdisciplinary field. Despite the fact that basics of the plasma phenomenon have been known since the 19th century, growing scientific attention has been paid in recent years to the use of plasma in medicine. Three most important plasma-based effects are pivotal for medical applications: (i) inactivation of a broad spectrum of microorganisms, (ii) stimulation of cell proliferation and angiogenesis with lower plasma treatment intensity, and (iii) inactivation of cells by initialization of cell death with higher plasma intensity. In this review, we explain the underlying chemical processes and reactive species involvement during NTP in human (or animal) tissues, as well as in bacteria inactivation, which leads to sterilization and indirectly supports wound healing. In addition, plasma-mediated modifications of medical surfaces, such as surgical instruments or implants, are described. This review focuses on the existing knowledge on NTP-based in vitro and in vivo studies and highlights potential opportunities for the development of novel therapeutic methods. A full understanding of the NTP mechanisms of action is urgently needed for the further development of modern plasma-based medicine.

## 1. Introduction

### 1.1. What Is the Plasma?

Plasma, often called the fourth state of matter, is actually an ionized form of gas. It contains neutral atoms, electrons, and ions, as well as excited atoms and radicals. However, not every collection of mobile gaseous particles can be called a plasma. They must be electrically quasi-neutral, exist in an excited state, and electric charges must be properly concentrated. In general, plasma is nothing like the other three states of matter. It consists of a ’great zoo’ of different chemical and physical entities, creating a very complex system of ions or mixed molecules interacting chemically and physically [1,2,3,4]. The physicochemical characteristics of plasma depend on a broad spectrum of parameters, including the type and composition of the gas or gas mixture used for plasma generation, the applied energy and electrode configuration, and the pressure [5].

### 1.2. Types of Plasma

The plethora of different plasma types, called by many different names and abbreviations, has been designed and described [6,7]. With regard to its application in the medical context, useful classifications are thermal plasmas (TP) versus non-thermal plasmas (NTP) and low-pressure versus atmospheric pressure plasmas [6,8] (Figure 1). Surgical techniques using such plasmas have been firmly established for a long time, even if they were not explicitly referred to as plasma medicine at the time.

The temperature of thermal plasma starts from about 10^6^–10^7^ K. It is generated in special reactors by accelerating electrons between two electrodes in a gaseous environment. Collisions between electrons and molecules increase the temperature by transfer of kinetic energy [9]. However, for direct application in living tissue, as in the main objective of plasma-assisted medicine, only plasma generated under mild temperature and atmospheric pressure conditions should be used. NTP, which is the main topic of this review, uses an electric field as an excitation source. Due to the type of reactor and the discharge used, there are many sources of NTP generation, including dielectric barrier discharge (DBD), glow, and corona or gliding discharge (Figure 1). Regardless of the NTP generation technique, its main advantages are low cost, the controllability of the process, and the easy switch on–switch off ability [2,7,10].

The atmospheric pressure plasma jet (APPJ) represents an improvement on DBD devices, which can normally only be used in enclosed devices. Moreover, in the case of DBD the surface is required to be a part of the high voltage electrical circuit. These problems do not apply to plasma jet devices. The main purpose of the APPJ is to extend the plasma plume over a greater distance. This allows remote processing of samples [11,12]. The concept was first introduced in 1998, and has recently become one of the most widely discussed and widely used types of plasma [12].

### 1.3. Reactive Forms in Non-Thermal Plasma

NTP differs from thermal plasma because of the properties of electrons, ions, and neutrality. Reactive forms of oxygen (reactive oxygen species, ROS), reactive nitrogen species (RNS), charged particles, neutral species, ozone, and ultraviolet (UV) radiation have all been found in NTP, making it a feasible tool for various applications. Although cold plasmas are a source of potentially toxic reactive species, their presence can induce a hormesis effect; thus, proper use of these ionized gas sources, with adapted and controlled conditions can bring about beneficial effects on the treated cells and tissues. Regarding their interaction with biological systems, the most important seem to be reactive oxygen and reactive nitrogen species (Table 1, Figure 2) [6,13].

ROS are extremely reactive molecules that can exist with one or more unpaired electrons. They play essential roles in the functioning of biochemical processes in the human body, known under the common term ‘redox signaling’. However, the elevated formation of different ROS leads to oxidative stress and consequently to activation of oxidative stress response; any imbalance in these processes is followed by molecular damage [14]. H_2_O_2_ is one of the predominant ROS varieties, a long-lived form which can affect cells for a long time after plasma exposure (Figure 2). Additionally, it is able to diffuse, and as such can influence matter farther away from the application site [15].

Reactive forms of nitrogen are various nitric oxide-derived compounds (Figure 2), including nitroxyl anion, nitrosonium cation, higher oxides of nitrogen, S-nitrosothiols, and dinitrosyl iron complexes [16]. They possess pleiotropic properties on cellular targets after both post-translational modifications and interactions with reactive oxygen species. Elevated levels of RNS augment the effects of ROS, and consequently have been implicated in cell injury and death by inducing nitrosative stress [17]. One of the most toxic RNS is ONOO^−^, which is formed from NO^.^ and O_2_^−^ and has been recently tested as a potential anti-cancer agent [16].

ROS and RNS can be classified into long-lived and short-lived species, as depicted in Figure 2. The lifetimes of primary reactive species are relatively short; thus, they can affect only the cells and molecules near the cold plasma source over a short time period (nano-, micro-, and millisecond). Several of these reactive species immediately combine with neutrals, water molecules, and other reactive species, forming secondary reactive species. These long-lived forms can affect cells relatively far from the NTP source through diffusion, and act over a long time [18,19,20,21]. Among the most long-lived forms are hydrogen peroxide (H_2_O_2_) and nitrite (HNO_2_/NO_2_^−^), whereas singlet oxygen (^1^O_2_) and nitric oxide (NO) belong to the second group. Their potential cellular targets are lipids, proteins, and DNA. All these reactive forms are able to trigger, inter alia, lipid peroxidation, protein damage, and membrane disturbances, eventually resulting in cell death (see Table 1) [17]. The exact mechanisms of reactive species influence in selected NTP applications are discussed in the next paragraphs.

## 2. Plasma Antiseptic Properties

### 2.1. Sterilization and Decontamination

The most commonly described and earliest discovered application of plasma in medicine is sterilization. Sterilization plays a major role in preventing the spread of disease in healthcare facilities. Current sterilization methods include autoclave, ethylene oxide gas, hydrogen peroxide gas plasma, and radiation. Plasma, due to its extraordinary antiseptic properties, can be applied to both surfaces and equipment in medical centers, as well as directly to wounds. Plasma application effectively destroys bacteria, viruses, and fungi, making its use highly versatile [22,23].

Plasma has been used for surface sterilization since the 19th century, when it was first applied to purify biologically polluted water. The basis of the method at that time was the use of ozone generated by dielectric barrier discharge plasma (DBD) [24,25]. The first patent for plasma sterilization dates back to 1970 and concerns the use of argon plasma, which was gaining popularity at the time, to clean the interior of vials [25]. Since then, there has been a plethora of scientific research and papers describing the microbiocidal properties of plasma and elucidating different mechanisms of microorganism inactivation (recently reviewed in [7]). Studies have proven that plasma inactivates bacteria not only at the point of application but in the area around it as well; see Figure 3 [26]. Although the recently accumulated knowledge has led to improvements in the efficiency of disinfection and sterilization using plasma-based technology [27], further investigations are required to fully understand the underlying mechanisms involved [24].

Recent augmentation of plasma sterilization was developed by Mackinder et al. [28] who tested the impact of magnetic effect on sterilizing action of oxygen and argon mixture plasma. It turned out that the magnetized plasma significantly reduces the time of effective sterilization. In addition, they showed that the plasma at the discharge point has a very high density, which generates high temperatures, that can damage medical surfaces.

Unfortunately, many bacteria produce endospores or other persistent forms that are resistant to many biocidal agents, both chemical and physical. *Bacillus subtilis*, which exists in food products, especially in bread, is an example of such a bacterium. As a non-pathogenic bacterium, it has been extensively used as a surrogate microorganism for studies instead of pathogenic bacteria. Mendes-Oliveira et al. [29] proved that cold plasma generated by dielectric barrier discharge was able to inactivate bacteria cells and their spores. As a result of the plasma action, the reactive gas, which in this case was ozone, led to a decrease in spore survival. In another set of experiments, it was proved that DBD can also inactivate *Escherichia coli* and *Vibrio parahaemolyticus* on plastic surfaces [30].

Patil et al. [31] investigated mortality of *Bacillus atrophaeus* in sealed polypropylene containers. The published set of results proved that atmospheric cold plasma discharge at a high voltage level inactivates spores within seconds both in a closed package and in the air. *Bacillus cereus* spores are one of the most popular contamination bacterial spores which can be inactivated by atmospheric-pressure plasma jet in dry environment [32]. In this case, the main killing factor for the bacteria was the interaction of radicals and charged particles with reactive ozone. As the moisture level in the reaction environment was increased, the chemistry of the plasma changed. Other strong oxidants, such as ROS/RNS, acted as the main activating agents, replacing ozone.

Bearing in mind that increased humidity changes the chemistry of plasma, the possibilities of water decontamination through plasma treatment have been investigated. On the other hand, studies in various liquids have proven that the sterilization effect depends on the solvent. In the case of physiological saline, it is possible to observe a significant decrease in bacteria cell numbers after plasma treatment; however, in the case of phosphate-buffered saline solution no such decrease was observed. While the influence of pH can be assumed, the reason seems to be more complex. The differences in bacteria inactivation mechanisms in these two different environments are attributed to the reactive species created by a complex system of plasma-induced chemical reactions in the liquid phase [25].

Plasma-based decontamination techniques are very comprehensive because they are able to remove contaminants from water, wastewater, and air [33,34]. The process of removing contaminants from water during plasma treatment is mainly ascribed to reactive species such as OH·, O_3_, NO_2_^−^, NO_3_^−^, ONOO^−^, H·, and O_2_^−^, free radicals, UV, and electric fields. It has been proven that NTP is able to remove chemical pollutants with efficiency close to 100%, The yield of the process depends on the type of plasma used, and the introduction of this method to the industry still requires a great deal of research [35].

### 2.2. Hemostasis and Wound Healing

Another aspect of using plasma-based techniques for sterilization is its application directly to human tissues. For this, it is extremely important to develop a method of producing a low-temperature plasma that does not cause pain and burning at the site of application. Laroussi et al. [36] proved that this is possible by using, e.g., helium as a carrier gas.

Essentially, the majority of chronic wounds are associated with aging and age-related diseases, such as diabetes, chronic inflammation, or other civilization diseases [37]. At present, medical personnel must deal with ever more hard-to-heal wounds for which treatment is a major challenge, despite the enormous development of medicine [38].

Wound healing in the human body is achieved through four main stages: hemostasis, inflammation, proliferation, and remodeling. In addition to the innate ability to regenerate, there are numerous factors that influence the entire process or its individual phases [37,38]. The first and most crucial issue in plasma-assisted wound healing is to achieve the proper effect on blood coagulation (Figure 4). Thermal plasma which has been used in cauterization devices causes a local temperature rise, and in consequence it cauterizes and closes the blood vessels; however, it damages the surrounding tissues as well. Formerly, this method was widely used; now, however, it is possible to apply NTP and trigger the positive influence on blood coagulation without heating [6,39].

#### 2.2.1. In Vitro Studies

The literature review shows that there are numerous reports proving that NTP accelerates blood clot formation [6,39]. The mechanisms of the process are not thoroughly known; presumably, plasma can interfere with biochemical reactions taking place during clotting, modify the activity of involved molecules through reactive species, act directly on abundant proteins agglomeration, and promote activation of platelets and fibrin filament formation (Figure 4) [39,40]. Plasma is able to speed up coagulation from 15 min to under one minute; an exposure of 15 s was reported as sufficient for coagulation [41].

Fridman et al. [26] used a floating DBD electrode to generate plasma and test how it affects the blood clotting rate. It turned out that after contact with the plasma the blood coagulated faster than in the control sample. Moreover, low doses of plasma not only augment the natural coagulation process, they have antiseptic properties, thereby reducing the risk of infection while not causing tissue damage.

Bacteria inactivation is one of the most important aspects of wound healing. The biofilm of chronic wounds consists mainly of such bacteria species as *Staphylococcus*, *Pseudomonas*, *Peptoniphilus*, *Enterobacter*, *Stenotrophomonas*, *Finegoldia*, and *Serratia*. Their presence is the main reason for the occurrence of infections in patients who suffer from chronic wounds and bedsores. Harnessing the biocidal properties of plasma in the healing process of infected wounds was investigated by Nasir et al. [42] The interactions between DBD (and CGDC) and the most common bacteria strains isolated from infected wounds, namely, methicillin-resistant *Staphylococcus aureus* (MRSA) and *Pseudomonas aeruginosa* (*P. aeruginosa*), were studied [43]. DBD exhibited a larger healing area, and after 1 h exposure almost all of the MRSA colonies were killed. Regarding *P. aeruginosa* cells, the effective treatment time could be shortened to 10 min, however, the use of DBD generated higher power consumption. On the other hand, the high efficacy observed with CGCD in treating chronic wound bacteria can be underpinned with the higher electron temperate and higher rotational temperature.

The anti-infective and wound-healing activity of plasma are not only ascribed to its chemical properties; it turns out that plasma triggers changes in extracellular matrix (ECM) proteins on the gene expression level [44]. Expression of genes involved in wound healing, such as type I collagen, transforming growth factors (TGFβ1/2), and alpha-smooth muscle actin (α-SMA), have been reported to be modified by plasma exposure. Undoubtedly, this issue deserves further elucidation.

#### 2.2.2. In Vivo Studies

Development of blood coagulation methods using NTP conditions is crucial for numerous surgical applications. Ferri et al. [45] used argon plasma to optimize tonsillectomy in children. It turned out that replacement of the traditional “cold” dissection by “hot” argon plasma coagulation (APC)-assisted tonsillectomy significantly reduced the operative time and intraoperative blood loss without increasing the postoperative morbidity. Moreover, the benefits of this method include low cost and uncomplicated procedures. Another in vivo experiment relied on 15 s plasma irradiation of the cut saphenous or tail vein in mice [41]. This study proved the positive impact of plasma on blood coagulation, which is very important because venous hemorrhages are often difficult to stop, can be a huge threat, and may even lead to death.

The positive effect of NTP on wound healing via anti-inflammatory properties has been well demonstrated in vivo by Hung et al. [46] Sprague-Dawley rats were used as the test subject, with 2 × 2 cm wounds cut on their backs. The experimental group was exposed to plasma for five minutes a day for four weeks. NTP was introduced diffusely into the cage where the rats were housed. Comparison with the control sample showed that the plasma treatment shortened the healing time without damaging internal organs. In addition, rats that were not exposed to plasma developed inflammation, infection, and exudation. An elevated level of white blood cells was indicative of the inflammatory process, whereas 4-hydroxy-2-nonenal (4-HNE) served as reactive oxygen species marker. The latter was increased in the NTP-treated group, confirming that ROS are engaged in the wound healing process.

The group of Chatraie [47] observed the effect of NTP on wound healing on the dorsal skin of adult rats. Under general anesthesia and sterile conditions, two circular magnets were used to create pressure ulcers. The wounds were divided randomly into control and plasma-treated groups. Animals in the plasma-treated group received plasma radiation for 5 days within 3 weeks (Figure 5), and there were no particular differences in the appearance of the wound surfaces between two tested groups; however, the mechanical strength of the repaired wounds in the plasma-treated group was significantly higher than that in the control group, and differences appeared in the wound size on particular days of the healing period. The wound size in the plasma-treated group was significantly smaller than in the control group from day 7 to day 21. The authors concluded that NTP provided an improvement in wound healing through promoting re-epithelialization, wound closure, late-phase inflammation, and tissue repair.

APPJ has found applications in wound healing as well, mainly due to a number of advantages over other plasma devices, including free gas source, long working distance, high gas flow rate, and larger area of effective treatment [48,49].

As mentioned above, the key aspect of wound healing is the antiseptic and disinfecting effect. It is important to decontaminate the entire wound surface. As a standard, the wound is irradiated with a single stream of plasma [50]. One of the types of Plasma Multi-Jets devices tested against various bacteria, including drug resistant strains, is a multi-spot plasma applicator which generates 52 mm-sized helium jets. It was modified to diffuse-mode plasma application, with additional plasma volume generated in between the 52 multi-jets. The possibilities of multi-stream applicators were successfully tested against both Gram-positive and Gram-negative bacteria such as *S. aureus*, *P. aeruginosa*, *E. coli*, *S. dysgalactiae*, and *B. subtilis*. This method remains in the initial phase of development, and clinical trials are required for its implementation; nevertheless, it holds great promise for the treatment of chronic and infected wounds. In addition, this method was effective against oral microorganisms such as *Lactobacillus casei*, *Streptococcus mutans*, *Candida albicans* and *Escherichia coli*, making it possible to use APPJ for the treatment of oral wounds and bacterial diseases [51,52,53].

Patients infected with the HIV virus are particularly vulnerable to the occurrence of hard-to-heal wounds. Eguiluz et al. [54] described the case of an infected patient who was diagnosed with a tumor in the neck area. The tumor was removed and NTP therapy was applied as a coadjutant of wound closure. Plasma was used during the operation to treat muscular tissue, subcutaneous cellular tissue, and skin. Studies have shown the positive effect of NTP on wound healing. NTP reduces the inflammatory phase, protein synthesis (collagen), and fibroplasia, and combats invading pathogens. Additionally, the patient in this case study reported only mild postoperative pain, which may indicate an analgesic effect of plasma.

### 2.3. Anti-Viral Activity

The inactivation of viruses is a very important topic, and is much less well understood than bacteria inactivation. The global COVID-19 pandemic has significantly increased interest in this issue. NTP was tested as a potential method for SARS-CoV-2 inactivation. The possible mechanism of antiviral activity is linked to active species that can damage the virus capsid, genome, and other proteins and glycoproteins [55].

During the COVID-19 pandemic, special attention was paid to hand hygiene and disinfection with the purpose of reducing resident skin flora and eliminating transient pathogenic flora. Research conducted by Daeschlein et al. [56] shows that APPJ has promising potential for skin disinfection. This opens new horizons, as alcohol-based skin antiseptics that are currently in use cause discomfort during application to pre-injured skin. In addition, they generate problems associated with handling, transporting, distributing, and storing inflammable liquids.

Moreover, plasma has shown promising results in face mask sterilization. It has been proven that viruses lose infectivity with a log-4 reduction when exposed to ozone gas produced by a DBD plasma generator. This is very important because to date there has been no available method for sterilizing masks without reducing their effectiveness. The ability to recycle face masks would result in lower production, less medical waste, and consequently less environmental pollution, and would be beneficial from an economic point of view as well. Results show it is possible to disinfect mask surfaces at least five times without compromising filtration efficiency [57].

Most important is the sterilization of the air, where the virus spreads the fastest; thus, it is worth conducting research aimed at improving this approach [58]. NTP efficiently decontaminated SARS-CoV-2-containing bioaerosols, surfaces, skin, and personal protective equipment. However, it can be difficult to set up NTP devices for large spaces. It is possible to inactivate coronaviruses in a water environment as well. In this process, S proteins are damaged by RONS generated in argon NTP. Plasma jets produce reactive species that dissolve in the liquid and interact with each other. In this way, tyrosine, tryptophan, and histidine within the Receptor-Binding Domain (RBD) are oxidized, which affects the ability of the virus to bind to the ACE2 cellular receptor.

## 3. Scarring and Skin Regeneration

When discussing the topic of wound healing, the scarring process cannot be avoided. Scars are often non-aesthetic remnants of wounds. Various methods of their removal, reduction, and leveling are constantly being developed. Most of the available approaches allow the treatment of scars only after they have completely healed. The use of cold plasma could support the scarring process already during wound healing. Metelmann et al. [59] observed scarring process in five patients over 10 days, 6 months, and 12 months. The subjects were treated with argon plasma and the applied method showed great improvement in aesthetic values from the beginning to the end of scar formation.

As mentioned in the Introduction section, plasma gas composition is of decisive importance in many practical applications. Usually, active carriers such as oxygen and nitrogen or a neutral gas, typically argon, are used. Other reactive environments can be formed in situ, which has been reported by, e.g., Vasilets et al. [60]. They described the action of an NO-containing gaseous mixture, that was generated as a reaction product between ammonia and oxygen, ${4NH}_{3}+{5O}_{2}\vec{\leftarrow }4NO+{6H}_{2}O,$ and applied for treatment of skin scars, wounds, and articular musculoskeletal disorders. Such changes in the oxidative properties of the atmosphere drastically affected the treatment of abnormal skin scars and excessive skin scarring, as well as supporting the healing of ulcers. It was hypothesized that the observed beneficial therapeutic responses may be attributed to the synergistic action of NO with other plasma gas species.

Rad et al. investigated the effect of surface DBD, an Ar plasma jet, and an He plasma jet. The produced plasmas were applied to skin wounds of BALB/c mice. It was experimentally proven that He plasma jet application resulted in the fastest wound healing, which lasted eleven days. These data show that plasma tailoring and control are essential to achieve a beneficial biological effect, and that the careful choice of plasma source influences the final effect [61].

Skin is the largest organ in the human body, and is an extremely important protective barrier against harmful factors in the external environment [50]. Any skin damage can become a threat to human life. The positive effect of cold plasma on the process of wound healing and scarring raises the further question of whether plasma can affect the condition and appearance of unwounded skin. Studies have shown that the use of plasma in dermatology not only improves wounds and ulcers treatment and wound contraction, it stimulates angiogenesis, promotes re-epithelialization, and fosters keratinocyte and fibroblast migration while inhibiting the proliferation of the hyperproliferative keratinocytes [62]. Several of these plasma-induced effects can be achieved through improvement of the skin oxygenation and a hormetic effect of mild oxidative stress. Oxygen partial pressure (pO_2_) is a key factor playing a role in cell and tissue physiology and pathology. The improved oxygenation may be ascribed to the effect of plasma temperature ranging from 30 to 40 °C, as the use of the term “cold plasma” does not mean that it is actually cold [50].

The most important condition for skin health is hydration, and plasma could find applications here as well. Fluhr et al. [63] observed a decrease in hydration of the dermis and an increase of transepidermal water loss after exposure to plasma. Tests were performed in vivo on human skin. In contrary, Athanasopolous et al. [64] proved that using helium plasma enhanced the wettability of human skin. The application of NTP on the human stratum corneum resulted in a more hydrophilic skin surface. The skin samples used in the study were excised from the breasts of a 63-year-old woman. The treatment lasted 30, 60, 90, 120, or 150 s, then the water contact angle was measured. Very low values were obtained, proving the excellent wettability of the skin.

Another important parameter of the skin is its pH. The physiological pH of skin is slightly acidic, ranging from 4 to 6 depending on the part of the body and individual predisposition [65]. It is widely accepted that plasma has acidifying properties due to the action of acids formed from reactive nitrogen forms [66]. Lowering the pH value can have both positive and negative effects. It supports the healing of skin damage and accelerates regeneration by stimulating fibroblast proliferation. Acid peelings have been used in cosmetology and aesthetic medicine for a long time. They have a mild exfoliating effect, helping to fight acne and minor scars. Properly used, they show excellent clinical performance and have a great effect on skin rejuvenation [67]. However, the main disadvantage of exfoliating acids is their allergenicity. The use of plasma does not generate this problem. If on-going research can prove whether the use of plasma is able to induce similar to acids effects of healing and regeneration, such NTP-based treatments could become more widely available.

## 4. Anti-Cancer Therapies

Cancer is a civilization disease posing a serious threat to public health. According to the report of the international agency for cancer research, the number of cases reached over 19 million in the year 2020. The number of deaths was calculated at up to 10 million [68]. The field of oncological medicine is still being developed, and research groups are racing to find newer and newer methods of cancer treatment; however, despite occasional successes treatment for most cancers is a long way from being satisfactory. Despite the huge milestones that oncology has achieved over the years and the implementation of many new treatments, an effective cure for cancer remains a Holy Grail in oncology. The most basic and frequently used clinical approaches are surgery, chemotherapy, and radiotherapy. However, there are cancers that are resistant to these therapies, while others are capable of recurring even after being temporarily cured [69].

Recently, a lot of research, both in vitro and in vivo, focuses on using NTP for cancer treatment and proving its efficacy. Low-power plasma can stop the proliferation and progression of cancer cells, and higher power can even lead to their death. The general anticancer mechanism of NTP is based on the activities of reactive oxygen and nitrogen species [70,71]. Many reactive species among those appearing in NTP are toxic to cancer cells according to the same mechanisms indicated in Table 1.

### 4.1. In Vitro Studies

Several in vitro studies have shown that NTP exhibits anticancer activity against various cancers, including these that are resistant to other therapies, such as glioblastoma or melanoma [72]. Specifically, combining NTP with conventional therapeutic methods is attracting growing attention as a potential novel treatment approach [70,73]. Walk et al. [74] investigated the influence of cold plasma on neuroblastoma cells. In vitro treatments were performed at 4 kV in a helium flow, and neuroblastoma cells were treated for 0, 30, 60, and 120 s. Cytometric analysis was performed immediately after irradiation as well as 24 and 48 h and 7 days after treatment. Measurements of the percentage of apoptotic cells taken immediately after exposure did not show a positive result. After some time, however, the pro-apoptotic effect of NTP was observed. The hypothesis explaining this effect is that the presence of ROS leads to oxidative stress and induces cell apoptosis, and can additionally cause senescence or cell cycle arrest [74].

Pancreatic cancer is one of the most dangerous cancers. When a tumor is surgically removed, cancer cells (so-called positive margins) are left behind in the patient’s body. Special plasma applicators could help to destroy the remnant microscopic amount of tumor cells intra-operatively. In vitro studies have shown an increase in the rate of apoptosis as a time-dependent result of plasma exposure. In the case of 5 s of treatment a higher apoptotic rate was achieved after 48 h, while in the case of 10 s of treatment the effect was most beneficial after 72 h [75].

Recently, Shakya et al. [76] compared the effectiveness of APPJ and DBD techniques during in vitro studies against different cancer cell lines. Direct treatments (applied directly to growing cells) and indirect treatments (involving interactions with culture media), were related. For all tested cell lines like lung cancer (A549 and SW900), melanomas (B16-F16 and G361), glioblastomas (U251SP and U87MG), and breast cancer (MDA-MB-231 and MCF-7), the induction of apoptosis was observed independently on the kind of treatment. Summing up, the collected results show that regardless the device type, duration, and mode of treatment, NTP can play a vital role in reducing cancer cell viability and inducing apoptosis. A diverse variety of cancer cell lines have been successfully subjected to APPJ treatment, reflecting its usefulness and opening up new possibilities around its potential future applications.

The effectiveness of indirect action of plasma-activated solutions in destroying cancer cells was confirmed in a study using human neuroblastoma SH-SY5Y cells, murine myoblast C2C12 cells, and murine embryonic fibroblasts. The cells were exposed to plasma-activated culture media. The results of the study suggested that reactive oxygen species (ROS) generated during exposure of the medium and two kinds of saline to NTP act on the cells and trigger various effects, e.g., loss of translation inhibitor Pdcd4 and inducing regulated but atypical cell death different from apoptosis and necrosis/necroptosis [77]. In addition, plasma-mediated modification of currently available medicinal substances or biologically active molecules has been studied in terms of anti-cancer activity. Controlled plasma-assisted alterations of lysozyme led to a large number of oxidized amino acids, which in turn resulted in greater distortion of the lysozyme structure. Plasma-treated lysozyme has been shown to induce apoptosis of cancer cells [78]. Furthermore, structural modifications of cytoglobin have been reported to impair its protective activity against NTP-induced oxidative stress in melanoma cells [79].

Possible applications of APPJ in cancer research are currently being widely explored, and have even managed to overshadow DBDs. The effectiveness of APPJ in cancer treatment creates promising perspectives [80].

### 4.2. In Vivo Studies

Vandamme et al. [81] performed a pioneering study involving mice to investigate the therapeutic effect of plasma on brain glioma. The mouse population consisted of athymic females. Primary xenografts were obtained by subcutaneous injection of a suspension of human malignant glioblastoma cells into the nude animal’s hind legs. The mice were divided into a study group undergoing plasma treatment and a control group. Dielectric barrier discharge plasma was used to irradiate tumor grafts, which were then imaged using bioluminescence. The results of the study were very promising, showing a significant reduction of 56% in glioma volume at the end of treatment along with an increase in the lifespan of the mice, with the median survival being 9.5 and 15.0 days for control and plasma-treated mice, respectively.

The NTP-mediated treatment of another type of cancer, neuroblastoma, was tested in mice by Walk et al. [74]. The above-mentioned in vitro studies on these tumor cells showed promising results that were further reflected in mouse model. During the in vivo study, a single treatment ablated the tumors, the eventual tumor growth rate was decelerated, and the overall survival of the mice nearly doubled.

In another study, NTP was used in a mouse melanoma model. It turned out that it can destroys cancer cells, as well as stimulate systemic anti-cancer immunity. As a result, NTP therapy caused prolonged survival of the animals [82].

The above-mentioned studies have laid the foundation for further clinical research, although there are unexpectedly few clinical trials concerning the anti-cancer activity of NTP. Notably, in vivo and in vitro testing of NTP-assisted treatments is needed in order to better understand the effects of plasma-generated ROS/RNS and other species on biomolecules biological liquids, and cells, including cancer cells and tumors [72,73].

### 4.3. Selectivity

A basic problem with most cancer therapies is their low selectivity, which causes damage to healthy cells as an inevitable accompaniment to death of tumor cells. Recently, several research reports have shown that NTP could be a solution to this problem [73,83]. Plasma exhibits considerably higher efficiency when applied to cancer cells than in the case of healthy cells [84].

This phenomenon may be explained by the expression of aquaporins. Aquaporins are integral membrane proteins that serve as channels in the transfer of water, and in certain cases of small solutes across the cell membrane. Cancer cells tend to produce more aquaporins than normal tissues. Therefore, hydrogen peroxide and other active species diffuse much easier and faster into cancer cells (see Figure 6). According to Keidar et al. [85], after treatment with NTP the H_2_O_2_ derived from it diffuses freely into cells, increasing the concentration of ROS in tumor tissues. An additional confirmation of this hypothesis may be the fact that a high removal rate of H_2_O_2_ from the glioblastoma medium was observed.

Another set of experiments by Keidar et al. [86] investigated the selectivity of plasma treatment in vitro and in vivo. Research has been carried out comparing normal human bronchial epithelial cells (NHBE) to lung cancer cells (SW900), murine melanoma cells, and primary bone marrow macrophages. Human bronchial epithelial and lung cancer cells were treated with cold plasma for 30 s. It was observed that most SW900 cells were detached from the culture plate and their quantity was reduced, a strong selective effect that was not observed in the case of NHBE. Similar results were obtained with mouse cells. Both murine melanoma cells and primary bone marrow macrophages were treated with plasma for 0, 30, 60, and 120 s. Flow cytometry analysis was performed 24 and 48 h after treatment. The analysis of the obtained results indicates a considerable treatment response in murine melanoma cells after both 24 and 48 h, whereas no changes were observed in the macrophage culture.

In vivo tests were performed with bladder cancer xenograft cells. Inoculated mice were divided into two groups: tested mice were treated with cold plasma, whereas the control group received no therapy. It was observed that plasma treatment led to effective tumor ablation, with neighboring tissues left unaffected. Moreover, tumor cells did not grow back or exhibit metastasis, which suggests that this method could be successfully applied for treating the positive margins of cancer tissue [86]. These experiments confirm that cold plasma can selectively ablate cancer cells without destroying normal cells.

## 5. Surface Modifications

A few years ago, plasma began to be used to modify surfaces and coatings used in medicine. This was a groundbreaking development, especially for implant technology, and new techniques are currently being developed. An undoubted advantage is that plasma-mediated processes influence the material surface only, leaving the properties of the bulk material unchanged. Moreover, there are many different possible interactions between plasma and surfaces, such as: deposition, etching, modification, functionalization, activation, and cleaning [24].

A great example of using plasma processes to modify surfaces is the production of Mg-enriched oxide coatings on titanium generated by plasma oxidation. The obtained materials have been proven to be bacteriostatic and fungistatic against *Staphylococcus aureus*, *Escherichia coli*, and *Candida albicans*, which suggests the possible application of Mg-doped titanium coatings as long-term implants dedicated to bone tissue [87]. Titanium implants modified with NTP have been demonstrated to boost the osteogenic differentiation of mouse MC3T3 pre-osteoblastic cells [88]. The expression levels of osteogenesis-related genes such as ALP, osteocalcin (Ocn), osteopontin (Opn), and runt-related transcription factor 2 (Runx2), as well as alkaline phosphatase (ALP) activity and matrix mineralization of mouse pre-osteoblasts, was augmented on NTP-modified surfaces.

Finke et al. [89] investigated plasma-modified coatings, comparing differently deposited amino-functionalized surfaces. They tested polymer coatings deposited on titanium alloy base and demonstrated that all of the plasma-modified materials promoted osseointegration, i.e., they adherence and “melting” of the growing bone cells together with the titanium structure. This research confirms that this type of material could be successfully used in bone implants.

Currently, 3D printing techniques, which can be used for medical tools as well, are attracting great attention. Due to potential contact with human tissues, these types of instruments have to be free of pathogenic microbes or other contaminants. With the aim of counteracting bacterial biofilm formation, Muro-Fraguas et al. [90] coated elements printed from polylactic acid with acrylic acid layers in an NTP environment. The plasma involved in the polymerization processes was generated using nitrogen, and the resulting surfaces were made with different numbers of acrylic acid coating passes (1, 2, 6, and 12, labelled as Ac1p, Ac2p, Ac6p, and Ac12p, respectively; see Figure 7). The biofilm formation of six different model bacteria strains on the prepared coatings, including *Pseudomonas aeruginosa* and *Staphylococcus aureus*, was evaluated.

These results prove that acrylic acid coatings provide the surface with greater hydrophilicity and lead to the formation of a hydration layer with a thickness related to the surface roughness. This hydration layer could explain the reduction of bacterial adhesion and prevention of biofilm formation. Coatings applied with 1, 2, or 6 passes exhibited the required antibiofilm activity for the majority of bacteria analyzed. The smaller the number of applied passes, the better the antibiofilm properties of the resulting coatings.

## 6. Summary and Conclusions

Despite the development of an enormous number of different medical treatments, human beings are failing in the battle with pathogens. Antibiotic-resistant bacteria, new viral infections, and therapy-resistant cancers force us to search for novel solutions. One such solution could be NTP, which is gaining increasing and deserved attention in the fields of medicine and biomedicine. Multitasking NTP, as a very promising and versatile tool for various applications, brings hope for the future. Direct and indirect plasma treatment, including but not limited to formation of RONS, can affect various physiological processes in cells and tissues, ranging from cell growth to cell death. However, there are many intermediate steps with different outcomes e.g., oxidative stress response, activation of defense mechanisms, anti-inflammatory processes, etc., that affect the global result. Our attempts to understand the detailed mechanisms and chemical interactions of these reactive species will open up new therapeutic possibilities. Despite extensive research on reactive species involvement, the specific molecular mechanisms underlying the effects of NTP on the gene and protein levels remain far from being fully understood. The above-mentioned difficulties may underlie the fact that there are unexpectedly few clinical trials concerning medical applications of NTP. Thus, further work is needed to reveal the full potential of NTP for clinical applications.

## Figures and Tables

**Figure 1 ijms-24-12667-f001:**
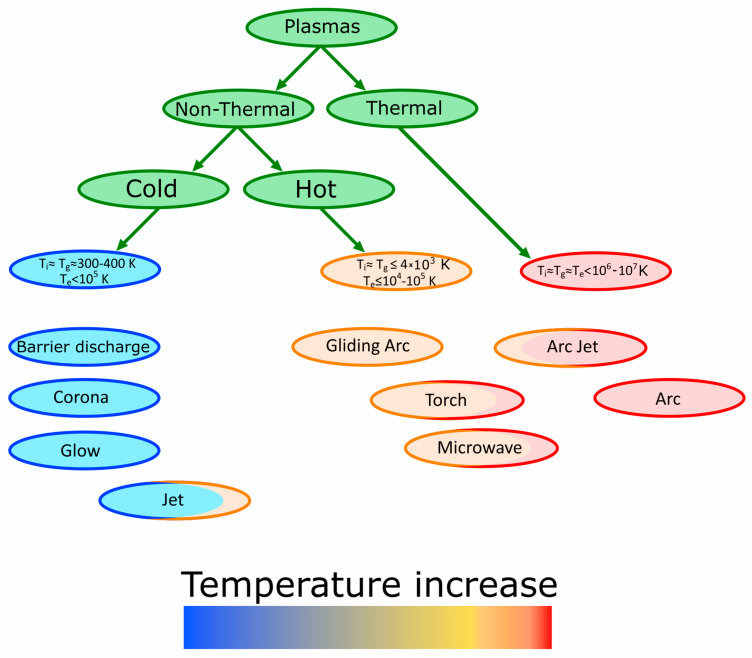
Plasma classification due to the temperature of the particles. Abbreviations: T_i_—ion temperature, T_e_—electron temperature, T_g_—neutral gas species temperature.

**Figure 2 ijms-24-12667-f002:**
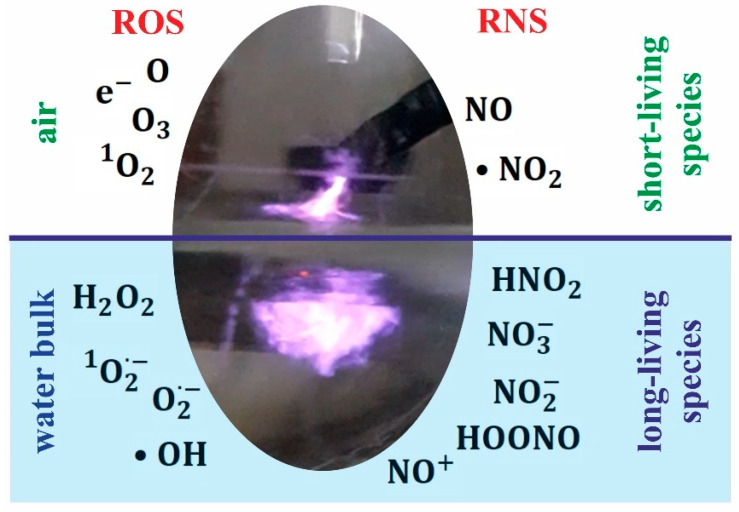
Main reactive species formed in NTP atmosphere. Picture taken 2s after switching on NTP (10 kV in Ar) over distilled water.

**Figure 3 ijms-24-12667-f003:**
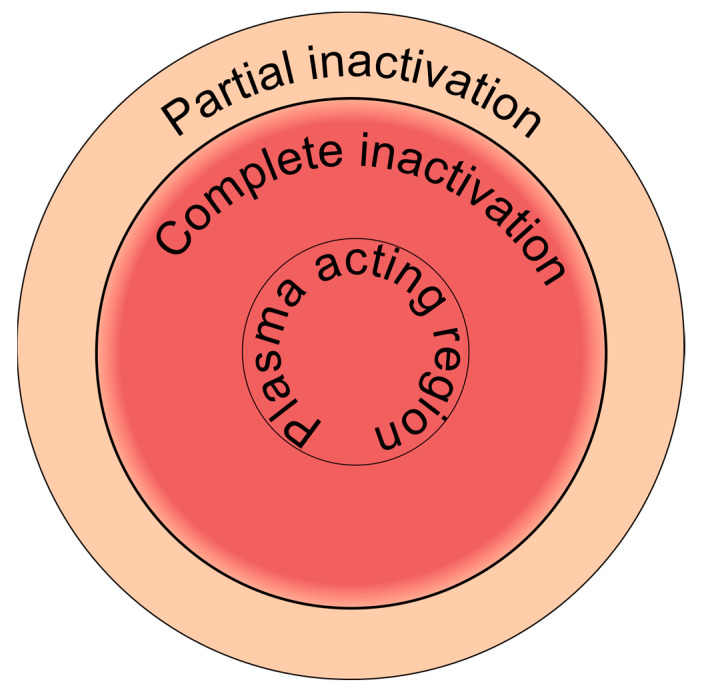
Schematic illustration of DBD effect on agar dishes with bacterial broth.

**Figure 4 ijms-24-12667-f004:**
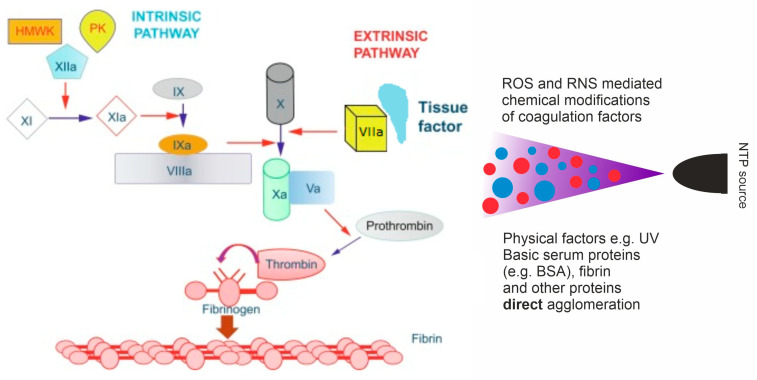
Simplified scheme of blood coagulation cascade influenced by NTP.

**Figure 5 ijms-24-12667-f005:**
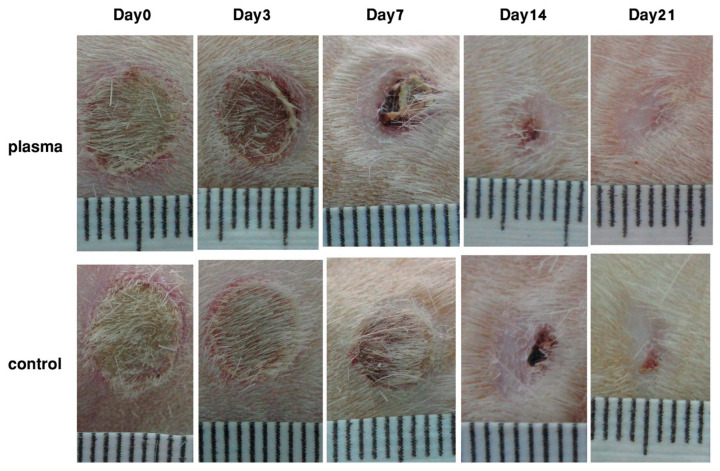
Macroscopic observation of wound healing on different days of healing period. Used with permission from [47]. Copyright 2023 Creative Commons license.

**Figure 6 ijms-24-12667-f006:**
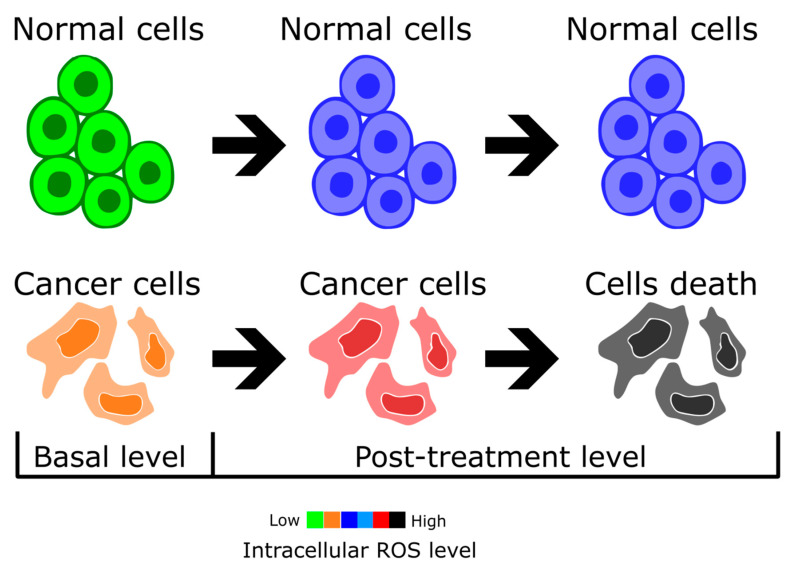
A scheme of selective ROS concentration and their effect on cell proliferation.

**Figure 7 ijms-24-12667-f007:**
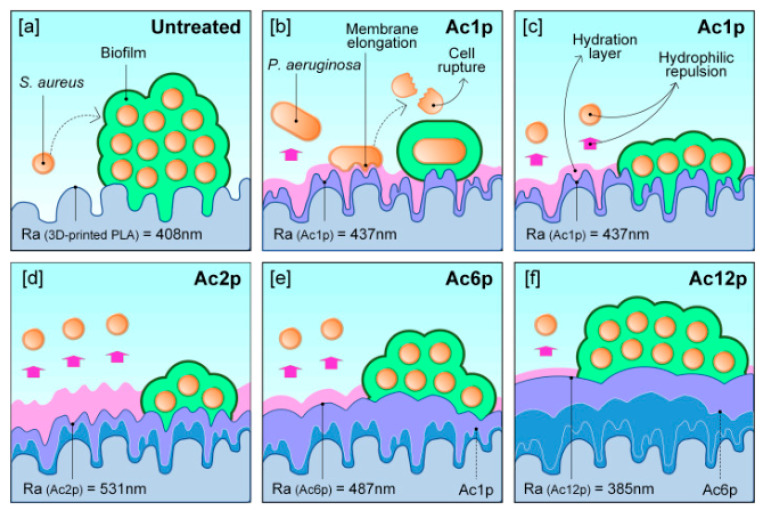
Scheme of bacterial biofilm formation on plasma-modified surfaces with different morphology and the consequent hydration layer: (**a**) untreated sample, (**b**) *P. aeruginosa* interaction with Ac1p (1 pass of coating), and (**c**) *S. aureus* interaction with Ac1p, along with *S. aureus* interactions with (**d**) Ac2p, (**e**) Ac6p, and (**f**) Ac12p. Used with permission from [90]. Copyright 2023 Elsevier.

**Table 1 ijms-24-12667-t001:** Characterization of active species formed by NTP.

Lifetime	Reactive Form	Action
Short-lived	·OH	Destroying of many vital molecules, including DNA, RNA, proteins, and phospholipids
	ONOO^−^	Damaging the function of antioxidant enzymes and other molecules
	^1^O_2_/·O_2_^–^	Cell proliferation or tissue regeneration
Long-lived	H_2_O_2_	Oxidation on many proteins and DNA
	NO_2_^–^/NO_3_^–^	Increasing of ONOO^−^ production
	O_3_	It has a genotoxic and cytotoxic effect
	NO	Causes oxidative stress, disrupted energy metabolism, DNA damage, activation of poly(ADP-ribose) polymerase, or dysregulation of cytosolic calcium

## Data Availability

Not applicable.

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
