# Peer review of "Non-Thermal Plasma Application in Medicine—Focus on Reactive Species Involvement"

_ijms, 2023, doi:10.3390/ijms241612667_

Round 1

Reviewer 1 Report

Thanks to the authors for submitting the review to the IJMS!

The use of non-thermal plasma (NTP) in medicine is a dynamically developing interdisciplinary field of research. In this review, in a very superficial, almost popular science format, the main chemical processes observed in the use of NTP for treating oncological and dermatological diseases and in the inactivation of bacteria are presented. There is a description of the plasma modification of medical surfaces such as surgical instruments or implants. All in all, the article deserves to be published, but only after substantial revisions to increase the number of sources analyzed and to develop the conclusions. In its present form, the article is a compilation of material that brings to light facts that are well known. The field in question has been in existence for several centuries and deserves a full review rather than a short review based on the analysis of 47 sources. The section on the discussion could be greatly expanded. Prospects for developing the direction and theoretical and application aspects of various limitations of the application of NTP could be detailed.

-

Reviewer 2 Report

Over the past decade, plasma medicine has become a dynamically developing interdisciplinary field of research. Every year, many new scientific achievements are made available in the form of publications. Each review should be a summary of the current state of knowledge. This publication is a concise study, but only selected aspects of the use of NTP in medicine. In addition, the sections are not written with the same scientific integrity, for example - chapter 3, the microbiological part is also described very broadly.

Comments
1. You should verify the work topic to the actual content
2. complete the literature with the more current  publications most references in this paper refer to publications before 2020
3. Correct the substantive level of individual chapters and their division into sections

4. Ver. 106-107 is controversial. The effect of NTP on the chemical composition of products has been verified, in many cases it is preferable

5. Ver. 281-282 is it definitely part of the publication section? 

6. Consider concentrating on a specific section of medicine in which NTP is used.

Reviewer 3 Report

Review on Non-thermal plasma application in medicine - short review

I have completed my review of manuscript IJMS-2521307, entitled, Non-thermal plasma application in medicine - short review.”

Atmospheric plasmas have recently made significant strides, leading to the emergence of nonthermal plasma. In recent decades, various innovative plasma diagnostic techniques have been employed to better understand the physics of nonthermal plasma. Nonthermal plasma is a groundbreaking method for generating reactive oxygen and nitrogen species. Plasma technology has numerous applications, including medicines.

The subject of the review is interesting and useful but the manuscript is not worthy of publication in present form. Substation major revisions required:

Comments for authors

Comment 1: To improve the abstract, it is suggested to add a statement highlighting the significance and novelty of this review, as it is not currently evident from the abstract. Additionally, it would be beneficial to end the abstract with a concluding statement to provide a clear summary and the importance of this review of what research gap is covered by this manuscript.

Comment 2: As the field of plasma is broad, the background information provided by the authors may be insufficient. It would be informative for readers if the authors included information regarding nonthermal plasma and its applications in the background section. The following recent article could be a useful resource to incorporate into the background information. [Review on the Biomedical and Environmental Applications of Nonthermal Plasma. Catalysts 2023, 13.]

Comment 3: Why authors choose to write a short review with a title covering a wide range of fields “Non-thermal plasma application in medicine.”

Comment 4: Nothing new is provided from lines 21 – 72 including Figures.

Comment 5: Why did the authors specifically focus only on DBD in Figure 4 to address bacterial inactivation? It is important to consider multiple plasma sources for both direct and indirect treatments in order to provide a comprehensive review and make a comparative information or statement to give something new to readers.

Comment 6: Could you please clarify the significance of Figure 5 and its connection to the role of plasma? If it primarily illustrates the pathways of the blood coagulation cascade, it may not be relevant to include it in this brief review.

Comment 7: I find Figure 6 to be lacking scientific rigor and its intended message unclear. It is difficult to comprehend what the authors intended to convey with this figure.

Comment 8: Figure 9 seems to present fundamental information without adequately incorporating plasma, which is the main focus of this review. It would be beneficial for the authors to include comments or an illustration specifically highlighting the role of plasma in the context of this review.

Comment 9: Rewrite the whole conclusion, there is nothing new provided in this paragraph.

Comment 10: The statement in the conclusion that suggests the molecular mechanisms underlying the effect of NTP on cell physiology are still poorly understood could potentially be subject to objection. Numerous researchers have made diligent efforts to shed light on the deeper mechanisms and gain a comprehensive understanding.

Comment 11: Only 47 references were provided for a review article, and most of the cited references are outdated. Recent articles are encouraged to cite.

Comment 12: The purpose of this short review remains unclear, and the selected title is overly broad. The manuscript lacks novel findings, and the authors' discussion regarding the pivotal role of reactive species in medical applications is inadequate. To provide the authors with an opportunity for improvement, it is recommended that they submit a revised version of the manuscript with comprehensive details, focusing primarily on the role of reactive species in medical applications. Additionally, a comparative analysis of different reactive species should be included to allow for comparative statements. The authors should take special care when designing the figure to encompass all aspects of plasma's role. Alternatively, the authors can consider either modifying the title to address a specific target area suitable for a short review article or expanding the manuscript to a full review.

Comment 13: The paper contains errors and typos that make it difficult to understand and distort its intended meaning. If the abbreviation is already proposed no need to repeat its full form in the whole manuscript. This is also the most repeated mistake. I encourage authors to reread carefully and fix all technical and grammatical errors.

The paper contains errors and typos that make it difficult to understand and distort its intended meaning. If the abbreviation is already proposed no need to repeat its full form in the whole manuscript. This is also the most repeated mistake. I encourage authors to reread carefully and fix all technical and grammatical errors.

Reviewer 4 Report

In this review paper, the authors introduced several major applications of non-thermal plasma in medicine, including sterilization and decontamination, wound healing and scarring, dermatology, and cancer treatment. This paper indeed provides a broad overview for the general reader. However, some errors and organizational issues need to be addressed before meriting publication. Please consider the following comments:

1) Some sentences are missing citations. For example, in line 60-62, page 2: “Elevated levels of RNS augment the effect of ROS, and consequently have been implicated in cell injury and death by inducing nitrosative stress.”

2) In page 4-6, when introducing the sterilization and decontamination applications, the authors should consider making a table summarizing the major proposed mechanisms in plasma-assisted sterilization and decontamination applications.

3) For different applications mentioned in section 2, what were the operating parameters used for plasma generation? What types of plasma sources (ICP or CCP) were preferred? What could be the preferred electron/ion energy for different types of applications?

4) Line 145, page 6. Thermal plasmas are equilibrium plasmas with typical temperature > 10000 K, as was also indicated in Figure 3. They are typically found in nuclear fusion reactors and star and stellar objects. Thus, how could thermal plasma be used in biomedical applications?

5) In section 2, most of discussed plasma treatments are based on DBD structure, while another important structure, atmospheric pressure plasma jet (APPJ), are not reviewed. APPJ has various advantages and already has commercialized products such as kINPen (see, for example, https://iopscience.iop.org/article/10.1088/1361-6463/aab3ad/meta). The introduction of APPJ should be added to form a complete review.    

6) The authors should recheck the paper for grammar errors and misleading sentences. 

Minor correction on grammer error is required.

Round 2

Reviewer 1 Report

Thanks to the authors for the work done to improve the content of the paper. The paper may be published in its current form.

Author Response

Dear Referee, thank you

Reviewer 3 Report

I have received the revised version with some minor improvements, but the main concerns regarding this manuscript remain unchanged. Firstly, the authors should highlight the revised text in the manuscript to make it easier for reviewers to identify the changes they made. Otherwise, locating the specific modifications made by the authors becomes quite challenging.

Additional comments.
1. Ensure not to restate the full form after introducing an abbreviation. Use the provided abbreviation, such as 'NTP' instead of 'Non-thermal plasma'.

2. The inclusion of outdated references throughout all sections raises concerns, as discussions on recent advancements and current knowledge are crucial. For instance, in sections 4.1 and 4.2, the authors cited only two or three references from 2013 and 2011 to support their entire argument. This practice is not considered acceptable in scientific writing, where up-to-date and relevant references are expected to be utilized.

3. In section 2, the authors are strongly encouraged to incorporate recent advancements in non-thermal plasma (NTP) for virus inactivation, including the potential application against COVID-19. This represents a critical and novel aspect that should be addressed in this review.

4. The conclusion section requires enhancement, as it currently appears to primarily provide background information only. For a impactful contribution to the scientific community, the authors should provide a novel and meaningful message to readers, that stems from their comprehensive review of the subject.

These are my major concerns on the present form of the manuscript that need to be addressed preciously by authors otherwise this review is not suitable for publication.

Ensure not to restate the full form after introducing an abbreviation. Use the provided abbreviation, such as 'NTP' instead of 'Non-thermal plasma'. Reread the manuscript preciously and correct such errors.

Reviewer 4 Report

The authors made no revision to sereval key comments such as the operating conditions, preferred plasma sources, and APPJ applications. From my point of view, these aspects must be discussed and/or addressed in a review article. Thus, I cannot recommend the publication of this paper in its present form. 

Round 3

Reviewer 3 Report

This third version seems improved after substantial revisions. I recommend accepting the manuscript for publication.

Author Response

Dear referee, thank you 

Reviewer 4 Report

I thank the authors for adding APPJ contents into the review paper. My suggestion is to scatter section 6 and put the corresponding contents into section 1-5 where specific applications are introduced. The general introduction of APPJ can be relocated to section 1.2 where other NTP generators are discussed. 

The authors should recheck the newly added contents for grammar errors and misleading sentences.

Author Response

Dear Referee we corrected the text according to your suggestions.